# Profiles of Frailty among Older People Users of a Home-Based Primary Care Service in an Urban Area of Barcelona (Spain): An Observational Study and Cluster Analysis

**DOI:** 10.3390/jcm10102106

**Published:** 2021-05-13

**Authors:** Juan-José Zamora-Sánchez, Edurne Zabaleta-del-Olmo, Sergio Fernández-Bertolín, Vicente Gea-Caballero, Iván Julián-Rochina, Gemma Pérez-Tortajada, Jordi Amblàs-Novellas

**Affiliations:** 1Atenció Primària Barcelona Ciutat, Gerència Territorial de Barcelona, Institut Català de la Salut, 08007 Barcelona, Spain; ezabaleta@idiapjgol.org; 2School of Nursing, Universitat de Barcelona, 08036 Barcelona, Spain; 3Fundació Institut Universitari per a la Recerca a l’Atenció Primària de Salut Jordi Gol i Gurina (IDIAPJGol), 08007 Barcelona, Spain; sfernandez@idiapjgol.org; 4Universitat Autònoma de Barcelona, 08193 Bellaterra (Cerdanyola del Vallès), Spain; 5Nursing Department, Faculty of Nursing, Universitat de Girona, 17004 Girona, Spain; 6Escuela de Enfermería La Fe, 46026 Valencia, Spain; gea_vic@gva.es; 7Grupo de investigación GREIACC, Instituto de Investigación Sanitaria La Fe, 46026 Valencia, Spain; 8Department of Nursing, Faculty of Nursing and Podiatry, Universitat de València, 46010 Valencia, Spain; ivan.julian@uv.es; 9Atenció Primària de l’Àrea Metropolitana Nord, Gerència Territorial Metropolitana Nord, Institut Català de la Salut, 08916 Barcelona, Spain; gperezto.mn.ics@gencat.cat; 10Central Catalonia Chronicity Research Group (C3RG), Centre for Health and Social Care Research (CESS), Universitat de Vic-University of Vic-Central University of Catalonia (UVIC-UCC), 08500 Vic, Spain; jamblas@chv.cat

**Keywords:** care dependence, cluster analysis, frailty, home care, multimorbidity, primary health care

## Abstract

Background: The multidimensional assessment of frailty allows stratifying it into degrees; however, there is still heterogeneity in the characteristics of people in each stratum. The aim of this study was to identify frailty profiles of older people users of a home-based primary care service. Methods: We carried out an observational study from January 2018 to January 2021. Participants were all people cared for a home-based primary care service. We performed a cluster analysis by applying a k-means clustering technique. Cluster labeling was determined with the 22 variables of the Frail-VIG index, age, and sex. We computed multiple indexes to assess the optimal number of clusters, and this was selected based on a clinical assessment of the best options. Results: Four hundred and twelve participants were clustered into six profiles. Three of these profiles corresponded to a moderate frailty degree, two to a severe frailty degree and one to a mild frailty degree. In addition, almost 75% of the participants were clustered into three profiles which corresponded to mild and moderate degree of frailty. Conclusions: Different profiles were found within the same degree of frailty. Knowledge of these profiles can be useful in developing strategies tailored to these differentiated care needs.

## 1. Introduction

The aging of the world’s population is contributing to an increase in the number of people with multimorbidity and complex care needs; as a result, the condition of frailty is receiving greater international interest [1]. Frailty is characterized by a decline in the functioning of multiple physiological systems accompanied by an increased vulnerability to stressors [2]. Moreover, frailty determines a reduction in functional reserve and a higher risk of adverse outcomes such as increased mortality, disability and use of healthcare resources [3,4,5]. It is also age-related, progressive, dynamic and potentially modifiable–and, therefore, preventable [6,7]. Furthermore, the introduction of the concept of frailty in clinical practice allows the identification of complex care needs by differentiating between chronological and biological age.

Today, the definition of frailty remains questioned, but according to the World Health Organisation, it is defined as “a progressive age-related decline in physiological systems that results in decreased reserves of intrinsic capacity, which confers extreme vulnerability to stressors and increases the risk of a range of adverse health outcomes” [8]. The main measurement approaches are the Fried’s frailty phenotype [9] and Rockwood and Mitnitski’s accumulation of deficits model [10]. The frailty phenotype is based on the presence of three or more of the following criteria: involuntary weight loss, exhaustion, slow gait speed, poor handgrip strength, and sedentary behaviour. Its use may be more suitable for the identification of non-disabled older people at risk of adverse health outcomes [11]. Regarding the accumulation of deficits model, allows an understanding of the concept through the complex interaction of physical, psychological and social factors that lead to frailty, expressed as a frailty index (FI). This FI is a ratio of potential age-related deficits that include disease, disability and geriatric syndromes [10]. The FI may be a more useful tool than the frailty phenotype to establish the effectiveness of any intervention and to describe the health status trajectories over time [11,12]. A large number of FIs have been developed based on this model, among which we highlight the Frail-VIG index. It has a multidimensional approach and is based on a comprehensive geriatric assessment (VIG is the Spanish/Catalan abbreviation for ‘comprehensive geriatric assessment). The Frail-VIG index fits well with the biopsychosocial model of primary care and allows stratification of people into degrees of frailty, making it easier to set tailored therapeutic goals and therapeutic intensity [13].

However, we still found heterogeneity among the characteristics of the people classified under each degree of frailty. Therefore, identifying different profiles of people would contribute to gaining additional knowledge about specific characteristics, and consequently, further facilitate personalization of the care plan. A number of previous studies have explored this approach with a comprehensive assessment approach among older people living in the community. Lafortune et al. identify four profiles, incorporating the physical and cognitive dimensions [14]. Armstrong et al. identify seven relatively homogeneous profiles among people receiving rehabilitation at home, which differ in terms of characteristics such as age, sex, cognition and functional impairment [15]. In a more recent study, Looman et al. identify six clusters around physical, cognitive and social functioning [16]. Other authors explore patterns based on multimorbidity [17,18], which is defined as the presence of two of more chronic diseases or conditions [19].

However, profiles based solely on morbidity do not capture other aspects of people’s complex needs when considering at-home care. In this regard, multidimensional FIs include interrelated concepts of interest such as morbidity, frailty and dependence. On the other hand, morbidity can be considered as an antecedent of frailty and dependence as a consequence of it [20,21]. Dependence is defined as a person’s lack of ability to perform basic activities needed for daily living as a result of a decrease in functional capacity [8]. It is important to identify these concepts correctly [22], because whilst they are related, their management is different and can confuse interventions designed to prevent and mitigate frailty. In addition, no research has been found that analyzed people’s profiles from a broad perspective, incorporating different domains of frailty (biological, psychological, and social), together with multimorbidity and dependence. Finally, people living with frailty risk experiencing a decline in their quality of life. Previous research shows the negative association between frailty, multimorbidity and dependence with respect to quality of life [23,24,25] Therefore, interventions aimed at decreasing frailty could have the added benefit of improving quality of life.

For all these reasons we decided to carry out this study, the aim of which was to identify frailty profiles of older people users of a home-based primary care service, from a comprehensive perspective, using the Frail-VIG index which integrates the concepts of frailty (physical, cognitive and social), multimorbidity and dependence.

## 2. Materials and Methods

### 2.1. Study Design, Scope, Period and Subjects

We carried out an observational study and cluster analysis in two health care areas of the Institut Català de la Salut (Catalan Health Institute) in Barcelona, Spain, that serve a population of 51,121 users. This is a population with a medium-high socioeconomic level and a lower percentage of immigration than the mean for Catalonia (9.8% and 8.8% respectively, compared to 14.6% for Catalonia) and a high level of over-aging (26.8% and 28.8% of people over 65 respectively, compared to the Catalan mean of 19.4%) [26]. The study period was from January 2018 to January 2021.

Participants were all people included to under the home-care program during 2018.

This program included all those who could not go to the health center for reasons of health, physical condition, or their social situation or environment. Two general practitioners and four community nurses manage this program, which was also supported by a social worker and a nurse case manager. All variables were assessed at baseline, except mortality, which was evaluated at 12 and 24 months after baseline.

### 2.2. Measurement Variables and Instruments

Frailty was measured using the Frail-VIG index. Published in 2017 by Amblàs- Novellas et al. [27], this FI based on the comprehensive geriatric assessment, consists of 22 trigger questions that are used to assess 25 deficits from eight assessment domains (functional, nutritional, cognitive, emotional, social, geriatric syndromes, severe symptoms and chronic diseases). The final score, which is obtained by dividing the accumulated deficits by the total potential accumulable deficits, range from 0 to 1 (with the submaximal limit in the clinical practice being close to 0.7). There is an Excel calculator available at: https://en.c3rg.com/index-fragil-vig (accessed on 11 May 2021). It enables people to be grouped according to degrees of frailty: non-frailty (<0.20), mild frailty (0.20–0.35), moderate frailty (0.36–0.50) and severe frailty (>0.50). The Frail-VIG index has shown a high prognostic value for 12-month mortality (value under the ROC curve of 0.9, 95% CI 0.88–0.92) [13,27].

For the cluster analysis, the variables were grouped as: sociodemographic (age, sex), diseases (cardiac, respiratory, renal, neurological, hepatic/digestive, cancer), symptoms (pain, dyspnoea), physical frailty (nutrition, falls, ulcers, dysphagia), psychological frailty (delirium, cognitive impairment, depression, anxiety), social vulnerability (the person lives alone or the family has limitations or difficulty in offering support, conflictive family relationship, inadequate hygiene conditions, inadequate housing, and lack of economic resources), and functional dependence for activities of daily living (ADLs) and instrumental activities of daily living (IADLs)

As a covariable, the perception of quality of life was measured by means of the EQ-5D-3L index, using the reference values for the Spanish population [28]. We also determined whether or not all participants had died at 12 and 24 months after the baseline assessment date, by reviewing primary care electronic health records, which systematically include the date of death.

### 2.3. Data Collection and Sources of Information

Community nurses from the home-care service performed a comprehensive geriatric assessment as part of their usual practice. They also administered by interview the two measurement instruments: the Frail-VIG index and the EQ-5D-3L index. The research team held a preliminary consensus meeting with these nurses to unify criteria for collecting the data from the aforementioned instruments, and a pilot test was carried out with 20 participants to identify potential problems and introduce strategies for improvement. After the pilot, no changes in the procedure were necessary, and an instruction manual was written to ensure homogeneous data collection.

For the assessment of mortality at 12 and 24 months after the baseline assessment, we reviewed electronic health records, which include up-to-date information on date of death, regardless of the place where the death occurred.

### 2.4. Study Size

No sampling strategy was used in this study. All the older people users of the home-based primary care service were included. In Spain, access to primary care is guaranteed by law and home care is part of the primary care service portfolio [29]. There is a comprehensive, multidisciplinary home care program; the study population was identified from the patient lists for this population-based care program.

### 2.5. Data Analysis

Categorical variables were expressed as frequencies and percentages and continuous variables as mean and standard deviation. Differences between clusters for descriptive variables were evaluated using the chi-square test, ANOVA and Kruskal–Wallis based on application conditions.

The cluster analysis was performed by applying the k-means hard clustering technique to the dataset. Prior to the clustering, a manual selection of variables was done to avoid overestimation of some data, and only the 22 variables of the frailty index, age and sex were used to determine the cluster labeling. The criterion for selecting these variables was to avoid duplicity of variables with similar clinical meaning. After grouping the individuals into different clusters with these features, all variables were used to characterize the clusters.

PCAmix compression technique was used to reduce dimensionality of the data, noise and deal efficiently with both dichotomous and continuous variables. Using this technique, principal component analysis (PCA) was applied to the continuous age variable and multiple correspondence analysis (MCA) to the dichotomous variables. According to Kaiser criterion, only the most significant 13 transformed dimensions were retained to feed the k-means clustering.

To assess the optimal number of clusters (k), Calinski-Harabasz, Xie-Beni and Silhouette indexes were computed from the average of 100 different realizations for each possible number from two to 20 [30]. To mitigate the negative effects of the random nature of the clustering, a clusterboot technique was used to select the most stable cluster from 100 different realizations. The Calinski index did not reveal clear evidence for any clusters, while Xie-beni and Silhouette gave better results for groups of three, six and nine clusters. Finally, the determination of the optimal number of clusters was defined based on the principle of parsimony and the potential clinical value of these three best options mentioned above. A cluster was considered clinically valuable when it facilitated more tailored care (targeted interventions) for frail people.

R version 3.6.3 software (Vienna, Austria) was used to preprocess the data and to apply the clustering and IBM SPSS Statistic version 24 (Armonk, NY, USA) to characterize the different obtained profiles.

### 2.6. Ethical Aspects

Informed consent was requested from the participants after they had been given oral and written information about the aims of the study. The study complies with Spanish regulations regarding the protection of personal data and was approved by the IDIAP’s Clinical Research Ethics Committee–registration number P17/150.

## 3. Results

A total of 412 people participated in the study, mostly women (68.4%), with a mean age of 88 (SD 8.1). Table 1 shows the characteristics of the participants in the total population. 31.3% of cases had three or more target organ diseases–the most prevalent being heart disease (60.2%) and chronic kidney failure (50.0%). More than 60% of participants reported significant functional dependence for ADLs and IADLs. Approximately half had some cognitive impairment, 51.5% showed social vulnerability and 17.0% lived alone. Quality of life was negatively correlated with the degree of frailty. We described this finding in more detail in a recent publication [31]. Overall mortality of the population at one year was 31.1% and 46.1% at two years, with significant differences between the six profiles identified (Table 1).

Comparing the three, six and nine cluster groups, the six-cluster group was found to be the most parsimonious and clinically valuable. Table 1 and Table 2 summarize the main characteristics of the six profiles identified, ordered by degree of frailty.

Profile 1 (25.7%). Mainly women with mild frailty, social vulnerability, moderate multimorbidity and dependence for activities of daily living. This was the most numerous profile. The mean age of participants in this profile was 87.7 and they were primarily female (76.4%). It was the only profile with mild frailty and the one with the lowest prevalence of cognitive impairment (12.3%). However, we highlighted social vulnerability among the majority (78.3%) and the higher percentage of cases living alone (40.6%) compared to the other profiles. 28.3% of the cases had three or more chronic diseases in target organs, the most prevalent being cardiac (55.7%) and renal (52.8%). This profile had the least dependence in regard to ADLs and IADLs (moderate-low) and the least accumulation of geriatric syndromes, with falls (25.5%) and chronic wounds (9.4%) being significant. Mortality at one year (13.2%) was the lowest of the six profiles identified.

Profile 2 (1.0%). People with moderate frailty affected by advanced cancer. This was the least numerous profile, as it only applies to four participants. Nevertheless, we decided to include it because of its clinical significance. The mean age was 83.8, and half were women. The participants were moderately dependent in regard to ADLs and IADLs. This group had the lowest social vulnerability, and all participants lived with someone. The degree of frailty was moderate, with a FI of 0.44. Mortality at one year was 100%.

Profile 3 (24.5%). Mainly people with moderate frailty, moderate dependence, cardiac disease and high multimorbidity. This was the second most numerous profile. The mean age was 86.1 and 56.4% of cases were women. The main characteristic of this profile was the high level of multimorbidity: half of the people had three or more chronic diseases in target organs, with the highest proportions in heart disease (84.2%), kidney disease (55.4%), respiratory disease (47.5%) and neurological disease (47.5%). It also had the highest rate of advanced chronic disease (17.8%). There was mild cognitive impairment among 23.8% of cases, and moderate-high dependence for ADLs and IADLs. 43.6% of cases were socially vulnerable, and 14.9% lived alone. A quarter of the participants had experienced falls and 12.9% had chronic wounds. The degree of frailty for this profile was moderate (FI 0.41). Mortality at one year was 29.7%.

Profile 4 (22.3%). Primarily nonagenarian women with moderate frailty, mild cognitive impairment and moderate-high dependence. This was one of the most numerous profiles, the oldest (a mean age of 92.7) and mainly women (84.8%). It was characterized by the high presence of mild cognitive impairment (83.7%) and moderate-high dependence for ADLs and IADLs. Social vulnerability was present in 39.1% of cases, with 6.5% living alone and 29.3% with a 24-h informal caregiver. 16.3% of the cases had three or more chronic diseases in target organs, the most prevalent being kidney (62.0%) and renal (40.2%) diseases. 21.7% of cases had three or more geriatric syndromes, with rates of delirium at 46.7%, falls 35.9% and ulcers 22.8%. All this implied a moderate degree of frailty (FI 0.39). Mortality at one year was 29.3%.

Profile 5 (15.8%). Mainly men with severe frailty, moderate-high multimorbidity and high dependence. The mean age under this profile was 87.9, and a minority were women (46.1%). They had a high proportion of moderate cognitive impairment (89.2%), and severe dependence for ADLs and IADLs. This was the second most socially vulnerable profile, at 53.8%. 7.7% of the people in this profile lived alone and 27.7% lived with an informal caregiver. 46.2% of cases had three or more chronic diseases in target organs, the most prevalent being cardiac (78.5%), neurological (61.5%) and renal (46.2%) diseases. Half of the people had three or more geriatric syndromes, with 56.9% having delirium. This profile had the highest proportion of falls (36.9%) and chronic wounds (36.9%). All of the above reflected a degree of severe frailty (FI 0.52). Mortality at one year was 52.3%.

Profile 6 (10.7%). Mainly women with severe frailty, cognitive impairment, dysphagia and total dependence. The cases in this profile were mainly women (77.3%), with a mean age of 84.4 years. We found the highest rate-93.2%-of cognitive impairment (moderate/severe) and total dependence for ADLs and IADLs. Social vulnerability was indicated in 29.5% of cases, and it was the profile with the highest percentage of permanent informal caregivers at 31.8%. This profile had the lowest burden of overall multimorbidity, with a predominance of neurological disease (52.3%) as well as over half the cases having three or more geriatric syndromes, with higher rates than under the other profiles for dysphagia (75%) and delirium (63.6%). Chronic wounds were also present in a third of cases. All of the above implied severe frailty (FI 0.53). Mortality at one year was 43.2%.

## 4. Discussion

The aim of the study was to identify frailty profiles among older people users of a home-based primary care service from a multidimensional perspective (physical, psychological and social). We identified six frailty profiles. Three of these profiles corresponded to a moderate frailty degree, two to a severe frailty degree and one to a mild frailty degree. In addition, almost three quarters of the study population were clustered into three profiles (profiles 1, 3 and 4) that corresponded to mild and moderate degree of frailty. It is they, therefore, who could benefit from planning frailty-prevention activities.

The profiles identified reveal underlying problems among the different domains of frailty and interrelated concepts (morbidity and dependence). These confirm the complexity of frailty, which is not sufficiently captured from measuring the degree of frailty and the distinction between the different domains affected (physical, psychological and social).

These results are in line with three previous studies that looked at profiles of people living in the community using a multidimensional assessment approach. Lafortune et al. [14] identify four profiles based on the assessment of physical and cognitive dimensions: relatively healthy, physical-cognitive impairment, significant physical impairment and significant cognitive impairment. Armstrong et al. [15] identify seven profiles among home-based rehabilitation clients, based on an assessment of physical, cognitive and social dimensions: people living alone and requiring assistance with housework and bathing; cognitively intact people who are independent in regard to ADLs; people requiring assistance with IADLs and bathing; women requiring assistance with IADLs; women requiring assistance with IADLs and some ADLs; people who are dependent but mobile and have cognitive problems; people who are dependent and immobile with cognitive problems. Looman et al. [16] identify six profiles based on physical, cognitive, and social functioning: relatively healthy; mild physical frailty; psychological frailty; severe physical frailty; medically frail; and multi-frail. All three studies identify a profile of relatively healthy older people, which did not appear in the present study. This could be related to the fact that the study population was composed of people included under a home-care program, who present greater vulnerability. In another recent study [32], three functional groups of older people at risk of disability are identified: physical, social and mixed. The psychosocial group, however, includes variables from the psychological and social dimensions, which can make it difficult to plan differentiated preventive or management activities.

The vast majority of frailty research that focuses on people’s outcomes is based on adverse outcomes, such as mortality, dependence, hospitalization and institutionalization. Outcomes related to positive aspects such as well-being and quality of life are less studied [24,33]. In our study, we have analyzed both negative outcomes (mortality, dependence, polypharmacy) as well as a positive outcome (quality of life). Frail older people can also present improvement in their quality of life. Therefore, the information will also serve to personalize the care plans based on a positive approach.

The main strength of this study is the statistical technique applied to determine the profiles, as well as the standardized data collection, using simple measurement instruments that are commonly used in healthcare practice. This pragmatic perspective would be feasible and capable of being extrapolated to larger populations. Another strong point is the differentiation between the concepts of frailty, morbidity and dependence, as it helps with planning care that is better aligned with needs. Where there is morbidity, activities will be planned to prevent or reduce frailty; where there is frailty, activities will be planned to prevent adverse outcomes, such as dependence [21].

### 4.1. Limitations and Future Research Recommendations

The main limitation of the study is that although the sample studied was large and all possible participants have been included, it corresponds to an aging population in an urban area with a medium-high socio-economic level, which makes it difficult to generalize the results to other settings or populations. Moreover, the Frail-VIG index is a simple and feasible instrument for use in the care setting, albeit with some limitations: the variables that correspond to physical frailty are focused on geriatric syndromes (weight loss, dysphagia, falls, ulcers) and do not include other variables that are recognized for identifying physical frailty. Here, the reference model would be Fried’s phenotype [9].

Future studies should contrast the results obtained in relation to other populations included under home-care programs with different socio-cultural levels and analyze other environments (for example, residential). It would also be useful to assess aspects of social frailty in greater depth. We suggest that this should include the three main components identified in a recent review: threat or absence of social resources to meet basic social needs; threat or absence of social behaviors and social activities; and threat or absence of self-management skills [34].

### 4.2. Care Implications and Outlook for the Future

The multidimensional nature of frailty has resulted in the identification of heterogeneous profiles, under which it would be useful to adapt care plans in health care practice. The multidimensional impact of frailty justifies multicomponent interventions [35,36,37]. Indications for multimorbidity management should also take account of health priorities, assess the potential benefits of interventions against potential harms, identify the person’s health trajectory, and encourage shared decision making, aligning people’s wishes and professionals’ opinions [1,38]. An effective overall management approach would include physical activity, nutritional supplementation, cognitive training, structured medication review, and social reinforcement. This approach shares similarities with other programs studied in our healthcare setting with the resources available in community care and other devices in the territory [39,40,41]. Table A1 in appendix shows, as an example a pragmatic proposal of care objectives and interventions for the different profiles identified, according to the current evidence for the management of frailty [35,36,37] and the multimorbidity [1,38]. Finally, it would also be helpful to include the opinions of the people cared for at home, their family members and caregivers on the profiles identified, as well as the care plans proposed for each of them.

## 5. Conclusions

This study identified six profiles of frailty, with clinical interest in older people users of a home-based primary care service. It shows that the population included under the home-care program is heterogeneous and has different care needs. Identifying these profiles complements information on degrees of frailty as it enables different care needs to be ascertained for people who are identified as having the same degree of frailty and is useful when designing interventions that are aligned with the needs of each population profile.

## Figures and Tables

**Table 1 jcm-10-02106-t001:** Characteristics of participants in each of the six profiles identified through cluster analysis. Values are absolute frequencies (percentages) unless stated otherwise.

Variables		Total * n* = 412	Profile 1 * n* = 106	Profile 2 * n* = 4	Profile 3 * n* = 101	Profile 4 * n* = 92	Profile 5 * n* = 65	Profile 6 *n* = 44	*p*-Value (*)
Age	Years, mean (SD)	88.0 (8.1)	87.7 (0.7)	83.8 (6.8)	86.1 (0.8)	92.7 (0.6)	87.9 (0.9)	84.4 (1.7)	<0.001
Sex	Men	130 (31.6)	25 (23.6)	2 (50.0)	44 (43.6)	14 (15.2)	35 (53.8)	10 (22.7)	<0.001
	Women	282 (68.4)	81 (76.4)	2 (50.0)	57 (56.4)	78 (84.8)	30 (46.2)	34 (77.3)	
Chronic Diseases	Cancer (active)	40 (9.7)	4 (3.8)	4 (100.0)	10 (9.9)	9 (9.8)	12 (18.5)	1 (2.3)	0.009
	Respiratory	116 (28.2)	29 (27.4)	3 (75.0)	48 (47.5)	7 (7.6)	20 (30.8)	9 (20.5)	<0.001
	Cardiac	248 (60.2)	59 (55.7)	0 (0.0)	85 (84.2)	37 (40.2)	51 (78.5)	16 (36.4)	<0.001
	Neurological	151 (36.7)	17 (16.0)	1 (25.0)	48 (47.5)	22 (23.9)	40 (61.5)	23 (52.3)	<0.001
	Digestive	39 (9.5)	16 (15.1)	0 (0)	7 (6.9)	6 (6.5)	6 (9.2)	4 (9.1)	0.234
	Renal (GFR < 60)	206 (50.0)	56 (52.8)	1 (25.0)	56 (55.4)	57 (62.0)	28 (43.1)	8 (18.2)	<0.001
	≥3 target organ diseases	129 (31.3)	30 (28.3)	1 (25.0)	51 (50.5)	15 (16.3)	30 (46.2)	2 (4.5)	<0.001
	Advanced Chronic Disease	32 (7.8)	3 (2.8)	4 (100)	18 (17.8)	1 (1.1)	5 (7.7)	1 (2.3)	<0.001
Severe Symptoms	Pain	87 (21.1)	26 (24.5)	1 (25.0)	34 (33.7)	20 (21.7)	6 (9.2)	0 (0.0)	<0.001
	Dyspnea	19 (4.6)	4 (3.8)	0 (0.0)	14 (13.9)	0 (0.0)	0 (0.0)	1 (2.3)	NA
Geriatric syndromes	Delirium	119 (28.9)	4 (3.8)	1 (25.0)	6 (5.9)	43 (46.7)	37 (56.9)	28 (63.6)	<0.001
	Falls	112 (27.2)	27 (25.5)	1 (25.0)	21 (20.8)	33 (35.9)	24 (36.9)	6 (13.6)	0.012
	Ulcers	85 (20.6)	10 (9.4)	2 (50.0)	13 (12.9)	21 (22.8)	24 (36.9)	15 (34.1)	<0.001
	Dysphagia	75 (18.2)	0 (0.0)	1 (25.0)	11 (10.9)	1 (1.1)	29 (44.6)	33 (75.0)	<0.001
	Incontinence	295 (71.6)	45 (42.5)	4 (100)	70 (69.3)	79 (85.9)	54 (83.1)	43 (97.7)	<0.001
	Polypharmacy (≥5 drugs)	365 (88.6)	92 (86.8)	4 (100.0)	100 (99.0)	74 (80.4)	60 (92.3)	35 (79.5)	<0.001
	≥3 geriatric syndromes	87 (21.1)	3 (2.8)	1 (25.0)	6 (5.9)	20 (21.7)	33 (50.8)	24 (54.5)	<0.001
Malnutrition	Weight loss ≥5% in the last 6 months	70 (17.0)	18 (17.0)	2 (50.0)	11 (10.9)	10 (10.9)	22 (33.8)	7 (15.9)	0.001
Cognitive impairment	Cognitive impairment	214 (51.9)	13 (12.3)	1 (25.0)	24 (23.8)	77 (83.7)	58 (89.2)	41 (93.2)	<0.001
	Pfeiffer (Range 0–10), mean (SD)	3.76 (3.33)	1.15 (0.15)	2.75 (1.80)	1.94 (0.20)	4.93 (0.23)	5.84 (0.36)	8.86 (0.39)	<0.001
Emotional Status	Treatment for depression	158 (38.3)	34 (32.1)	2 (50.0)	48 (47.5)	35 (38.0)	20 (30.8)	19 (43.2)	0.117
	Treatment for anxiety/insomnia	226 (54.9)	53 (50.0)	1 (25.0)	74 (73.3)	45 (48.9)	25 (38.5)	28 (63.6)	<0.001
Social Vulnerability	Presence of social vulnerability	212 (51.5)	83 (78.3)	1 (25.0)	44 (43.6)	36 (39.1)	35 (53.8)	13 (29.5)	<0.001
	Living alone	70 (17.0)	43 (40.6)	0 (0.0)	15 (14.9)	6 (6.5)	5 (7.7)	1 (2.3)	<0.001
	Professional caregiver 24 h a day	97 (23.5)	11 (10.4)	0 (0.0)	27 (26.7)	27 (29.3)	18 (27.7)	14 (31.8)	0.005
Basic Activities of Daily Living Barthel Index (BI)	Barthel Index, mean (SD)	48.4 (26.9)	73.0 (1.4)	35.0 (5.4)	51.1(1.7)	51.4 (1.8)	28.2 (3.3)	7.9 (2.1)	<0.001
	BI ≥ 95 No dependency	6 (1.5)	4 (3.8)	0 (0.0)	0 (0.0)	1 (1.1)	1 (1.5)	0 (0.0)	<0.001 (**)
	BI 90–65 (Mild–moderate)	143 (34.7)	85 (80.2)	0 (0.0)	26 (25.7)	22 (23.9)	9 (13.8)	1 (2.3)	
	BI 60–25 (Moderate–severe)	175 (42.5)	16 (15.1)	4 (100.0)	71 (70.3)	67 (72.8)	15 (23.1)	2 (4.5)	
	BI ≤ 20 Absolute dependency	88 (21.4)	1 (0.9)	0 (0.0)	4 (4.0)	2 (2.2)	40 (61.5)	41 (93.2)	
Instrumental Activities of Daily Living (needs help)	Money management	350 (85.0)	50 (47.2)	3 (75.0)	97 (96.0)	91 (98.9)	65 (100.0)	44 (100.0)	<0.001
	Telephone use	132 (32.0)	0 (0.0)	0 (0.0)	10 (9.9)	26 (28.3)	52 (80.0)	44 (100.0)	<0.001
	Medication management	304 (73.8)	19 (17.9)	2 (50.0)	85 (84.2)	89 (96.7)	65 (100.0)	44 (100.0)	<0.001
Frailty Frail-VIG index (FI) Range 0–1	Frail VIG index, mean (SD)	0.40 (0.12)	0.28 (0.01)	0.44 (0.02)	0.41 (0.01)	0.39 (0.01)	0.52 (0.01)	0.53 (0.01)	<0.001
	Non frailty (FI <0.20)	12 (2.9)	12 (11.3)	0 (0.0)	0 (0.0)	0 (0.0)	0 (0.0)	0 (0.0)	<0.001 (***)
	Mild frailty (FI 0.20–0.35)	116 (28.2)	68 (64.2)	0 (0.0)	23 (22.8)	24 (26.1)	1 (1.5)	0 (0.0)	
	Moderate frailty (FI 0.36–0.50)	191 (46.4)	26 (24.5)	4 (100)	63 (62.4)	58 (63.0)	28 (43.1)	12 (27.3)	
	Severe frailty (FI >0.50)	93 (22.6)	0 (0.0)	0 (0.0)	15 (14.9)	10 (10.9)	36 (55.4)	32 (72.7)	
Quality of life	EQ-5D-3L Index. (Range 0–1) mean (SD)	0.30 (0.23)	0.44 (0.18)	0.29 (0.16)	0.29 (0.21)	0.31 (0.24)	0.19 (0.24)	0.08 (0.01)	<0.001
Mortality	Mortality of 12 months	128 (31.1)	14 (13.2)	4 (100.0)	30 (29.7)	27 (29.3)	34 (52.3)	19 (43.2)	<0.001
	Mortality of 24 months	190 (46.1)	31 (29.2)	4 (100.0)	44 (43.6)	40 (43.5)	45 (69.2)	26 (59.1)	<0.001

(*) Profile 2 (4 cases) has been excluded in all *p*-value analyzes, in order to adjust to the application requirements of the tests. (**) The values of the Barthel index have been grouped into three levels (BI ≥ 95; BI 90–25; BI ≤ 20), to adapt the criteria for applying the chi-square test. (***) The frail-VIG index values have been grouped into three degrees of frailty (FI ≤ 0.35; FI 0.36–0.50; FI > 0.50) to adapt the criteria for applying the chi-square test. NA. The *p*-value calculation requirements do not apply. Abbreviations: BI, Barthel Index; FI, Frail-VIG Index; GFR, Glomerular Filtration Rate; SD, Standard Deviation.

**Table 2 jcm-10-02106-t002:** Summary of the variables used to define each of the six frailty profiles in the cluster analysis.

Profiles Domains	1(*n* = 106)	2(*n* = 4)	3(*n* = 101)	4(*n* = 92)	5(*n* = 65)	6(*n* = 44)
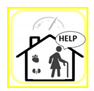	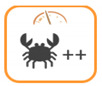	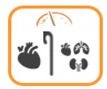	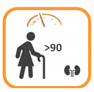	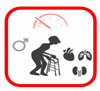	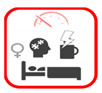
Degree of frailty	Mild frailty	Moderate frailty	Moderate frailty	Moderate frailty	Severe frailty	Severe frailty
Age (years)	87.7	83.8	86.1	92.7	87.9	84.4
Women (%)	76.4	50	56.4	84.8	46.1	77.3
Chronic diseasesand severe symptoms (0–6)	2	3.6	3.2	1.7	2.6	1.4
Physical frailty (0–4) (malnutrition, falls, ulcers, dysphagia)	0.6	1.6	0.5	0.7	1.5	1.4
Psychological frailty (0–5)(delirium, cognitive impairment, depression, anxiety/insomnia)	0.9	1.6	1.5	2.3	2.2	3.4
Social Frailty (0–1)(social vulnerability)	0.8	0.3	0.4	0.4	0.5	0.3
Functional dependencefor ADL,s and IADL,s (0–6)	1.8	3.3	3.7	4	5.2	5.9
Frail-VIG index (0–1)	0.28	0.44	0.41	0.39	0.52	0.53

The profiles are ordered according to the similarity between them. The darker the color, the higher frequencies or values of the variables. Frail-VIG index, higher scores represent a higher degree of frailty. Abbreviations: ADLs, Activities of Daily Living; IADLs, Instrumental Activities of Daily Living.

## Data Availability

The data presented in this study are available on request from the corresponding author. The data are not publicly available due to ethical reasons.

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
