# Peer review of "Profiles of Frailty among Older People Users of a Home-Based Primary Care Service in an Urban Area of Barcelona (Spain): An Observational Study and Cluster Analysis"

_jcm, 2021, doi:10.3390/jcm10102106_

Round 1

Reviewer 1 Report

The purpose of this study is to characterize the profiles of family among older adults of home-based primary care services. While the topic is important, several issues need to be addressed to enhance the quality of the study and clarification of methods. Here are my comments:

  1. Title: I found “the elderly” can be considered as a derogatory word in some cultures. “Older adults” is probably the term that is more accepted.
  2. Title: the source population is from Barcelona, Spain. To prevent overgeneralization of results, why not consider adding the city and country name in the title.
  3. Introduction: The authors should explain more explicitly about the gap in previous literature, which the current study is trying to fix. For example, the authors state that “However, profiles based solely on morbidity do not capture the aspects of people’s complex needs when considering at-home care”. Yet, the paragraph above did include some other aspects other than comorbidity. Thus, it is unclear what is the gap in identifying the profiles in previous studies.
  4. Introduction: Lines 83-86 on page 2 states that “The hypothesis for the study is that cluster analysis, …. “. A research hypothesis should be a statement on the proposed relationships between concepts. This statement does not specify any relationships and cannot count as a hypothesis. Thus, the authors may want to avoid mentioning hypothesis. In addition, k-means clustering is an unsupervised learning technique, which probably does not require a research hypothesis.
  5. Methods: The paper should include a statement on sampling strategy. Is it based on convenience sampling or any other type of sampling?
  6. Measurement: Lines 113-120 on Page 3, this is inappropriate use of semi-colons, given the components listed are not grammatically independent sentences. They should be replaced by commas.
  7. Measurement: This issue is partially related to the comment above regarding the selection of variables. More rationales for including these variables are needed. In addition, is there a reason to include the perception of quality of life since this was not mentioned in the introduction. Again, the introduction needs to have a more solid background to support the authors’ decision for including these variables.
  8. Data analysis: The authors should state which steps were implemented in R and which steps were implemented in SPSS for more transparency and reproducibility of data analysis.
  9. Results: In the results section, the authors mentioned “clinical significance” several times as a reason for certain analysis steps and reporting steps. However, the definition of “clinical significance” is unclear in this context. The authors should provide more explanation on what clinical significance mean.
  10. Results: The profiles of frailty row in Table 2 are hard to read. Instead of creating codes with footnotes, I think it might be much easier for readers if the authors could just spell out those labels.
  11. Results: The authors color-coded Table 2. However, the coding scheme is unclear.
  12. Limitations: limitation caused be sampling strategy should also be included, which affected the generalizability of findings.
  13. Implication: There seems to be a disconnection between findings and proposal in Table 3. The implication can be explained in more details.

Author Response

Response to Reviewer 1 Comments  (Manuscript ID: jcm-1182077)

We thank the reviewers and the editor for taking the time to assess our manuscript, and we greatly appreciate the thorough and thoughtful comments provide. Our point-by-point responses to the reviewers’ comments are attached below.

Point 1. Title: I found “the elderly” can be considered as a derogatory word in some cultures. “Older adults” is probably the term that is more accepted.

Response 1: We fully agree with the reviewer. We have modified the title in accordance with the reviewer’s suggestion. Likewise, we have substituted the term “elderly” for “older people” throughout the manuscript.

Point 2. Title: the source population is from Barcelona, Spain. To prevent overgeneralization of results, why not consider adding the city and country name in the title.

Response 2: Thank you for your comment. We have revised the title, and added the reviewer’s suggestion. The title of the revised manuscript is: “Profiles of Frailty among Older People Users of a Home-Based Primary Care Service in an Urban Area of Barcelona (Spain): an Observational Study and Cluster Analysis “

Point 3. Introduction: The authors should explain more explicitly about the gap in previous literature, which the current study is trying to fix. For example, the authors state that “However, profiles based solely on morbidity do not capture the aspects of people’s complex needs when considering at-home care”. Yet, the paragraph above did include some other aspects other than comorbidity. Thus, it is unclear what is the gap in identifying the profiles in previous studies.

Response 3:   We thank the reviewer for this comment. We have revised the introduction and expanded the information about the gap in identifying frailty profiles in previous literature.

Point 4. Introduction: Lines 83-86 on page 2 states that “The hypothesis for the study is that cluster analysis, …. “. A research hypothesis should be a statement on the proposed relationships between concepts. This statement does not specify any relationships and cannot count as a hypothesis. Thus, the authors may want to avoid mentioning hypothesis. In addition, k-means clustering is an unsupervised learning technique, which probably does not require a research hypothesis.

Response 4: We agree and we have revised the text of introduction. We have removed the text related to the hypothesis and modified the last paragraph of the introduction

Point 5. Methods: The paper should include a statement on sampling strategy. Is it based on convenience sampling or any other type of sampling?

Response 5:  We thank the reviewer for pointing this out. No sampling strategy was used in this study. The total number of people included in the home-based programme in the area during 2018 determined the sample size. We have explained this by adding a subheading in the methods section of the revised manuscript (please see subheading 2.4 Study size of the revised manuscript).

Point 6. Measurement: Lines 113-120 on Page 3, this is inappropriate use of semi-colons, given the components listed are not grammatically independent sentences. They should be replaced by commas.

Response 6: Thank you. We have fixed the error.

Point 7. Measurement: This issue is partially related to the comment above regarding the selection of variables. More rationales for including these variables are needed. In addition, is there a reason to include the perception of quality of life since this was not mentioned in the introduction. Again, the introduction needs to have a more solid background to support the authors’ decision for including these variables.

Response 7: We thank the reviewer for this comment. We have expanded the information on these aspects in the introduction. The definitions of the concepts of frailty, multimorbidity and dependence used in the study, the measurement approach and its relationship with quality of life have been included. We hope it is clearer.

Point 8. Data analysis: The authors should state which steps were implemented in R and which steps were implemented in SPSS for more transparency and reproducibility of data analysis.

Response 8: We agree and have revised the text to clarify it: the R software was used to preprocess the data and apply clustering and the IBM SPSS Statistic was used to characterize the different profiles obtained

Point 9. Results: In the results section, the authors mentioned “clinical significance” several times as a reason for certain analysis steps and reporting steps. However, the definition of “clinical significance” is unclear in this context. The authors should provide more explanation on what clinical significance mean.

Response 9:  We have expanded the information on these aspects in sections “data analysis” and “results”. (please see of the revised manuscript)

Point 10. Results: The profiles of frailty row in Table 2 are hard to read. Instead of creating codes with footnotes, I think it might be much easier for readers if the authors could just spell out those labels.

Response 10: We agree with the reviewer that the use of codes in the profiles in Table 2 is difficult to read. We have removed them and replaced them with the description of the degrees of frailty. We hope it is clearer.

Point 11. Results: The authors color-coded Table 2. However, the coding scheme is unclear.

Response 11: Thank you for your comment. We have improved the color by applying a scale of four-color intensities according to the value of each variable. We have included information on the interpretation of the color map of the variables included in Table 2 as well as the interpretation of the Frail-VIG index

Point 12. Limitations: limitation caused be sampling strategy should also be included, which affected the generalizability of findings.

Response 12:  We agree with the reviewer that it would be useful to address this point in more detail. We have expanded on this limitation in the revised manuscript. Please see subheading 4.1 Limitations and Future Research Recommendations of revised manuscript.

Point 13. Implication: There seems to be a disconnection between findings and proposal in Table 3. The implication can be explained in more details.

Response 13: We thank the reviewer for pointing this out. Although the content of table 3 is not the focus of this article, we consider that it would be interesting to have a practical example of the usefulness of the identified profiles, for care planning adapted to the needs of the people of each profile. We added the table as an appendix. The activities suggested in the appendix are based on current evidence for the management of frailty (Abbasi, 2018 doi:10.1503/cmaj.171509, Apóstolo, 2018 doi:10.11124/JBISRIR-2017-003382, Dent, 2019 doi:10.1016/S0140-6736(19)31785-4) and multimorbidity (Boehmer, 2018 doi:10.1371/journal.pone.0190852 Boyd, 2019 doi:10.1111/jgs.15809)

Reviewer 2 Report

Dear Authors,

I read with interest this Your manuscript, submitted for publication. I thank You for giving me this opportunity.  

Here are my comments and suggestions: 

1) As You wrote, definition of frailty is not unique. Indeed, some definitions are reported in published literature. I propose to insert in the Introduction section some of these, highlighting their main differences. A Table can be useful for the readers.

2) You used the Frail-VIG index. You should have reported what this index is, and not assume that the reader knew it. What You wrote in lines 103-106 was not enough.  

3) Morbidity, frailty and dependency should be better described (please, see lines 76-78).  Again, some concepts were taken for granted, but they should have been better described.  

4) You wrote "mortality was evaluated at 12 and 24 months after baseline" (line 101).  However, You evaluated mortality only at 12 months after baseline (please, see lines 193, 199, 210, 220, 230, and 241). 

5) As You correctly pointed out, Your results/conclusions may not be applicable to populations having low socio-economic levels. This should be specified in the Title: Profiles of frailty among elderly users of a home-based primary care service in an urban area with a medium-high socioeconomic level..... 

Author Response

Response to Reviewer 2 Comments (Manuscript ID: jcm-1182077)

We thank the reviewer and the editor for taking the time to assess our manuscript, and we greatly appreciate the thorough and thoughtful comments provide. Our point-by-point responses to the reviewer’s comments are attached below.

Point 1: As You wrote, definition of frailty is not unique. Indeed, some definitions are reported in published literature. I propose to insert in the Introduction section some of these, highlighting their main differences. A Table can be useful for the readers.

Response 1: We thank the reviewer for this comment. We have included the latest WHO definition of frailty in the World Report on Aging and health, and references to the two main frailty measurement operating models (please see second and third paragraph of the introduction of the revised manuscript)

Point 2: You used the Frail-VIG index. You should have reported what this index is, and not assume that the reader knew it. What You wrote in lines 103-106 was not enough. 

Response 2:  We have expanded on the information about the Frail-VIG index. (please see point 2.2. Measurement Variables and Instruments of the revised manuscript)

Point 3: Morbidity, frailty and dependency should be better described (please, see lines 76-78).  Again, some concepts were taken for granted, but they should have been better described.  

Response 3: Thank you for your comment. We have added your suggestions into the introduction. We have included the definition of the term frailty and the main models for evaluating frailty (please see response 1). Likewise, we have also included the definitions of multimorbidity and dependency the first time they appear in the text, in the introduction. (please see manuscript with marked changes)

Point 4: You wrote "mortality was evaluated at 12 and 24 months after baseline" (line 101).  However, You evaluated mortality only at 12 months after baseline (please, see lines 193, 199, 210, 220, 230, and 241). 

Response 4: In section “Results"(page 3), the mortality of the population at 12 and 24 months is indicated (line 178). The information on mortality at 12 and 24 months of the profiles is expanded in Table 1, referenced in line 179. The following paragraphs provide the global information of the profiles. Mortality is indicated only at 12 months, to avoid redundancy.

Point 5: As You correctly pointed out, Your results/conclusions may not be applicable to populations having low socio-economic levels. This should be specified in the Title: Profiles of frailty among elderly users of a home-based primary care service in an urban area with a medium-high socioeconomic level..... 

Response 5:  To avoid over-generalization of the results, we have considered adding the city and country name in the title, which was also suggested by the other reviewer. We hope it is enough to avoid an excessively long title: “Profiles of Frailty among Older People Users of a Home-Based Primary Care Service in an Urban Area of Barcelona (Spain): an Observational Study and Cluster Analysis “

Round 2

Reviewer 1 Report

  1. Last paragraph for introduction: You may want to specify the target population in your research purpose statement.
  2. It is still unclear why no sampling strategy is used. Is this because it is a census study, where authors collected data from every individual in source population? If so, the authors should explicitly state that. If not, please further explain.

Author Response

We thank the reviewer for taking the time to assess our manuscript, and we greatly appreciate the thorough and thoughtful comments provide. Our point-by-point responses to reviewer’s comments are attached below.

Point 1. Last paragraph for introduction: You may want to specify the target population in your research purpose statement.

Response 1: Thank you. We have specified the target population in the research purpose statement.

Point 2. It is still unclear why no sampling strategy is used. Is this because it is a census study, where authors collected data from every individual in source population? If so, the authors should explicitly state that. If not, please further explain.

Response 2: Thank you very much for your comment. We have expanded the information about sampling following the reviewer's recommendations.

Reviewer 2 Report

Dear Authors, I read with attention the newer, revised version of Your article. All my comments and suggestions were satisfactorily met. No doubt.

Author Response

We thank the reviewer for taking the time to evaluate our manuscript. We would also like to greatly appreciate the thorough and thoughtful comments he/she provided that helped to substantially improve the manuscript.